# Select forelimb muscles have evolved superfast contractile speed to support acrobatic social displays

Matthew J Fuxjager[1]*, Franz Goller[2], Annika Dirkse[1], Gloria D Sanin[1], Sarah Garcia[2]

[1]Department of Biology, Wake Forest University, Winston-Salem, United States; [2]Department of Biology, University of Utah, Salt Lake City, United States

**Abstract** Many species perform rapid limb movements as part of their elaborate courtship displays. However, because muscle performance is constrained by trade-offs between contraction speed and force, it is unclear how animals evolve the ability to produce both unusually fast appendage movement *and* limb force needed for locomotion. To address this issue, we compare the twitch speeds of forelimb muscles in a group of volant passerine birds, which produce different courtship displays. Our results show that the two taxa that perform exceptionally fast wing displays have evolved 'superfast' contractile kinetics in their main humeral retractor muscle. By contrast, the two muscles that generate the majority of aerodynamic force for flight show unmodified contractile kinetics. Altogether, these results suggest that muscle-specific adaptations in contractile speed allow certain birds to circumvent the intrinsic trade-off between muscular speed and force, and thereby use their forelimbs for both rapid gestural displays and powered locomotion.

*For correspondence: mfoxhunter@gmail.com

**Competing interests:** The authors declare that no competing interests exist.

## Introduction

Signal evolution in animals has produced a variety of unique and unusual displays, including the elaborate physical routines performed for social communication and advertisement (*Beehler and Pruett-Jones, 1983*; *Prum, 1990*; *Masonjones and Lewis, 1996*; *Voigt et al., 2001*; *Hogg and Forbes, 1997*). Little is understood about how these displays evolve, and their emergence represents a functional feat in animal design (*Fuxjager and Schlinger, 2015*; *Irschick et al., 2007*). Most physical displays, for example, involve rapid limb movements, which are controlled by the same neuro-motor architecture that governs the otherwise 'normal' movements used to power locomotion. However, because motor performance is constrained by a trade-off between muscle contraction force and speed (*Rome et al., 1999*; *Young and Rome, 2001*), limb muscles should in theory be unable to generate *both* the swift appendage movements incorporated in showy physical displays *and* the force needed to drive locomotion (*Young and Rome, 2001*; *Rome and Lindstedt, 1998*). How, then, do limb motor systems and their underlying musculature control extraordinarily fast limb movements necessary for the production of adaptive behavioral displays?

We study this issue in a group of volant passerine birds, which produce different types of elaborate courtship behavior (*Figure 1A*). Trade-offs between muscle force and contraction speed clearly exist in the wing muscles of birds within this size range (*Biewener, 2011*; *Biewener and Roberts, 2000*; *Biewener et al., 1992*; *Dial and Biewener, 1993*; *Hedrick et al., 2003*), and evidence suggests that such muscles maintain sufficient force-generating abilities to power flight and regulate flying speeds (*Biewener et al., 1992*; *Dial and Biewener, 1993*; *Hedrick et al., 2003*). Of the species we study, however, both golden-collared (*Manacus vitellinus*) and red-capped manakins (*Ceratopipra mentalis*) produce exceptionally rapid wing movements as part of their acrobatic courtship displays

**eLife digest** Many animals court mates and fight with rivals by performing physically elaborate and showy displays. From male fiddler crabs waving their claws to attract females, to the leaping dances of whooping cranes, these displays often involve remarkably fast limb movements. However, in many cases it is puzzling how animals can perform these behaviors, because the muscles that move the limbs are often geared to produce strength for walking, running or flying, and not speed. Indeed, decades of research in animal physiology has confirmed that limb-moving muscles can contract with either great strength or great speed, but never both.

A small group of tropical birds called manakins produce different types of courtship displays, including some in which the wings are moved extremely rapidly. To date, nobody has examined if or how the limb muscles can generate such superfast movements.

Fuxjager et al. now show that, in two species of manakins that produce rapid wing movements as part of their courtship displays, one of the main wing muscles has evolved to move the wings at superfast speeds. In fact, this muscle can move the wing at speeds that are more than twice as fast as those required for these birds to fly, and appears to be the fastest limb muscle on record for any animal with a backbone. Fuxjager et al. also show that the manakins' other wing muscles are no different from other birds, and suggest that these muscles are preserved to produce the strength needed for flying.

Further studies could now explore how this one muscle can create such superfast wing movements and whether male hormones, like testosterone, play a role in regulating the muscle's speed.

(*Bostwick and Prum, 2003*; *Fusani et al., 2007*; *Fuxjager et al., 2013*). For example, male golden-collared manakins perform *roll-snaps*, whereby they hit their wings together above their backs at ≈60 Hz to produce a loud mechanical sonation (*Fusani et al., 2007*; *Fuxjager et al., 2013*). Kinematic analyses suggest that this behavior is performed by elevating the opened (extended) wings and then by repeatedly retracting the wings to force the wrists to collide in quick succession (*Bostwick and Prum, 2003*; *Fusani et al., 2007*; *Fuxjager et al., 2013*). Likewise, male red-capped manakins produce a similar wing sonation called a *clap*, in which the wings are laterally extended slightly above the body and then immediately retracted back to the sides (*Bostwick and Prum, 2003*). *Claps* are also produced in rapid succession, with the wing-extension and wing-retraction phases occurring at ≈45 Hz (*Bostwick and Prum, 2003*). In both species, the wing oscillation frequencies used for courtship displays exceed those that similarly sized birds use for flight (≈25–30 Hz) (*Donovan et al., 2013*; *Pennycuick, 2001*); thus, the forelimb motor system is likely modified to actuate courtship movements, while preserving the force-generating ability needed to drive powered locomotion. However, the nature of these modifications has never been explored.

We hypothesize that, in both golden-collared and red-capped manakins, the main muscles involved in humeral retraction have evolved rapid contractile kinetics to support display maneuvering. This idea is rooted in the kinematic studies that suggest that retraction of the wings is a major feature of the *roll-snap* and *clap*. Building on this notion, we expect that the other primary wing muscles that supply the main force for forelimb movement during flight are unmodified. To test our hypothesis, we therefore measure twitch contraction frequencies in situ from the three main wing muscles: (*i*) the *supracoracoideus* (SC), which raises the wing by elevating the humerus; (*ii*) the *pectoralis* (PEC), which lowers the wing by depressing the humerus, and (*iii*) the *scapulohumeralis caudalis* (SH), which retracts the wing via the humerus [*Figure 1B*; *Biewener, 2011*, *Dial, 1992*, *Dial et al., 1991*]. Because the SH acts as the main wing retractor and generates the least aerodynamic force necessary for powered flight, we predict that its contractile kinetics are significantly increased in golden-collared and red-capped manakins, compared to the SC and PEC. In this vein, we predict that these latter two muscles show similar contractile kinetics, as they appear to contribute less to the rapid wing movements that are incorporated into these two birds' displays. Inasmuch, these muscles are likely more constrained by natural selection to generate aerodynamic force for powered flight (*Biewener et al., 1992*; *Dial and Biewener, 1993*; *Hedrick et al., 2003*).

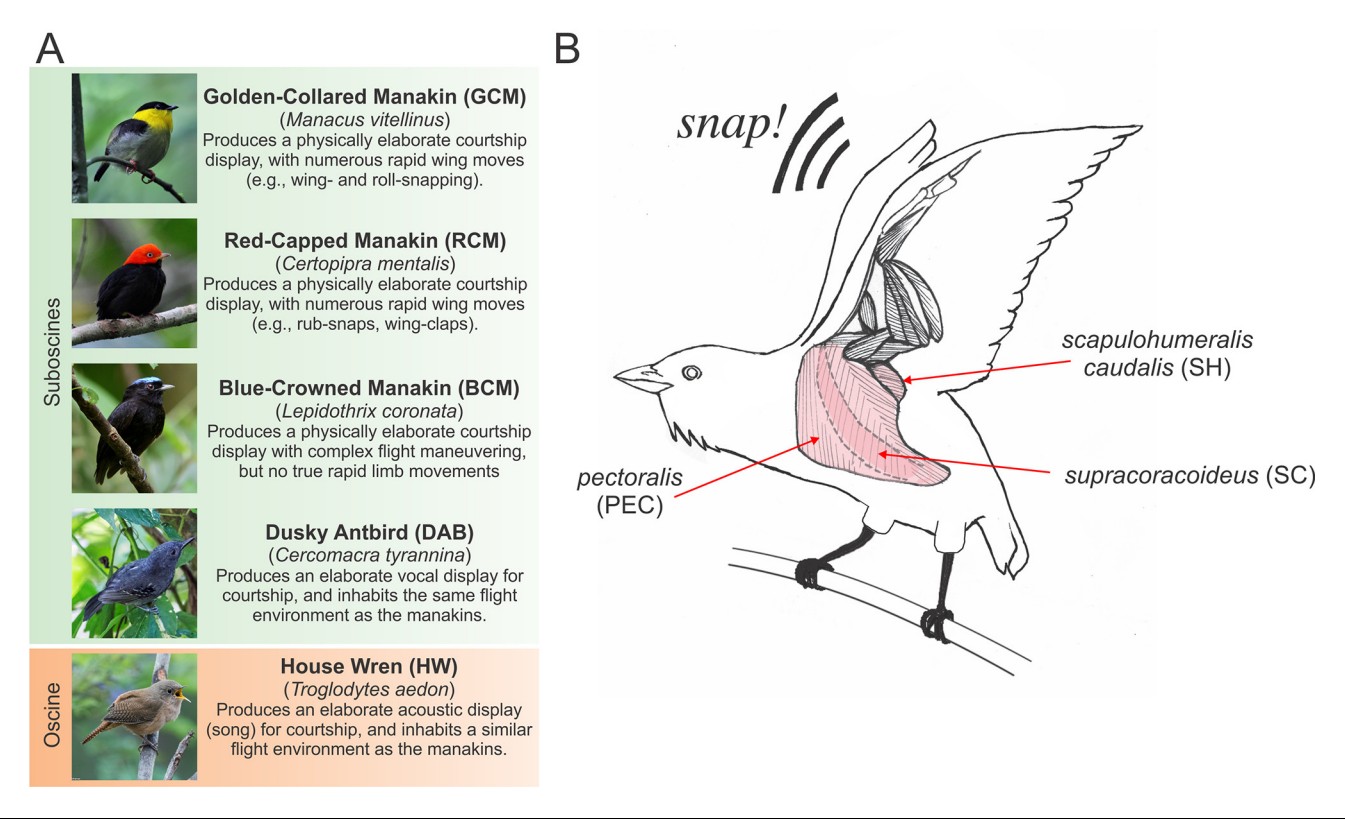

**Figure 1.** Species and muscles examined in this study. (**A**) Species included in our study, with common names in boldface typesetting and scientific names in italic typesetting. A brief description of each species' display and reason for inclusion in the study is described. Photos with permission from Nick Athanas. (**B**) Illustration of the three main wing muscles in a golden-collared manakin that are involved in the production of the *roll-snap*. These include (*i*) the *supracoracoideus* (SC), which raises the wing by elevating the humerus; (*ii*) the *pectoralis* (PEC), which lowers the wing by depressing the humerus, and (*iii*) the *scapulohumeralis caudalis* (SH), which retracts the wing via the humerus (*Biewener, 2011*; *Dial, 1992*; *Dial et al., 1991*). Note that the SC is a darker shade of pink, compared to the PEC and SH, because the SC lies deep to the PEC. Scientific illustrations of these muscles can be found elsewhere (*Welch and Altshuler, 2009; George and Berger, 1966*). This schematic is modified with permission from Schlinger, et al. (*Schlinger et al., 2013*).

To further examine whether unusually rapid muscle kinetics co-evolve with fast-moving wing displays, we compare muscle twitch frequencies from wild-caught golden-collared and red-capped manakins to those of three other related (wild-caught) species (*Figure 1A*). The first of these is the blue-crowned manakin (*Lepidothrix coronata*), which is a close relative of the aforementioned manakins and produces an elaborate physical display without rapid wing-snaps or wing-flicks (*Durães, 2009*). The second species is the dusky antbird (*Cercomarca tyrannica*), which inhabits the same niche as the manakins described above but produces elaborate vocalizations in lieu of a physical display (*Morton and Derrickson, 1996*; *Morton, 2000*). The last bird is the house wren (*Troglodytes aedon*), a more distant relative within the same avian order that sings to attract mates and does not produce a robust physical display (*Johnson and Kermott, 1991*). These comparisons collectively provide a framework for evaluating how the muscular properties that control twitch frequencies vary across species, and whether rapid wing maneuvering used for courtship is related to this variation. Therefore, based on our hypothesis described above, we predict that rapid kinetics of a humeral retractor muscle will be absent from blue-crowned manakins, dusky antbirds, and house wrens, as these species do not produce rapid wing maneuvers as part of their courtship displays.

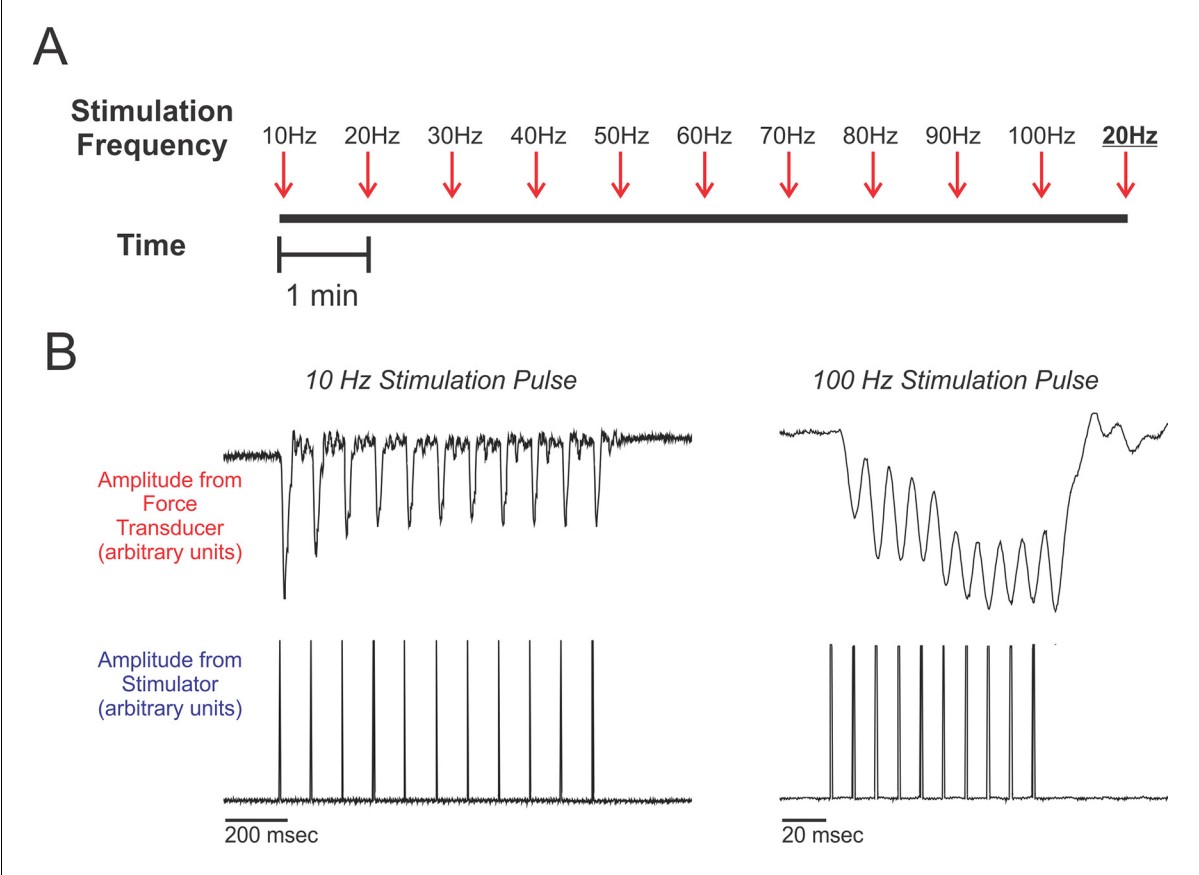

**Figure 2.** Experimental design. (**A**) Schematic of the work flow and procedural design. Muscles were prepared in situ (see methods) and stimulated at frequencies ranging from 10 Hz to 100 Hz, increasing at increments of 10 Hz. Stimulation trains were spaced 1 min apart. After the 100 Hz stimulation train, we administered a second 20 Hz stimulation train (shown underlined and in boldface typesetting). Percent recoveries were compared between this 20 Hz stimulation train and the first 20 Hz stimulation train to validate that the procedure did not exhaust/damage muscle. (**B**) Representative twitch recordings from a red-capped manakin (10 Hz from the SC and 100 Hz from the SH; note the differences in time scale). For each individual, percent recovery at a given stimulation frequency was calculated by averaging the percent recoveries of the first eight stimulations within the administered train. This corresponds to reasonable numbers of wing oscillations that golden-collared and red-capped manakins incorporate into their respective *roll-snap* or *clap* displays.

## Results

### Validation of in situ muscle twitch recordings

We measured mean levels of muscle relaxation in response to electrical stimulation at different frequencies (*Figures 2A,B*). Therefore, our first aim was to verify that the repeated stimulations administered to a given muscle did not exhaust or damage the tissue and thereby confound our results. To do this, we compared muscle recovery in response to the 20 Hz stimulation given at the beginning of our experimental series to muscle recovery in response to a second 20 Hz stimulation given at the end of our experimental series (*Figure 2A*). Overall, we found no significant difference in percent relaxation values between these two separate stimulation events (*Table 1*; all p values ≥0.18). These results therefore indicate that the muscles were functionally intact throughout the twitch frequency recording sessions.

A benefit of collecting muscle twitch recordings in situ is that we can evaluate whether the muscle produces movement in response to stimulation under normal load conditions (i.e., whether stimulated contractions can move the wing and thus overcome its weight and inertia). Indeed, we found that this was the case, as the muscle stimulations that we administered actuated wing movement. This included instances when we administered high stimulation frequencies that did not induce

**Table 1.** Mean (± 1 SEM) percent muscle relaxation in response to a 20 Hz stimulation at the beginning (first) and end (second) of the stimulation series. GCM = golden-collared manakin; RCM = red-capped manakin; BCM = blue-crowned manakin; DAB = dusky antbird; HW = house wren.

| Species | Muscle | First 20 Hz stimulation | Second 20 Hz stimulation | t Statistic | p Value* |
|---|---|---|---|---|---|
| GCM | PEC | 99.06 (0.94) | 99.95 (0.05) | -1.0 | 0.42 |
| | SC | 99.44 (0.49) | 99.61 (0.35) | -2.03 | 0.18 |
| | SH | 100 (0.0) | 95.27 (4.73) | -1.09 | 0.39 |
| RCM‡ | PEC | 100 (0.0) | 100 (0.0) | - | NS |
| | SC | 100 (0.0) | 100 (0.0) | - | NS |
| | SH | 100 (0.0) | 100 (0.0) | - | NS |
| BCM | PEC | 85.01 (14.98) | 100 (0.0) | -1.0 | 0.42 |
| | SC | 100 (0.0) | 99.90 (0.097) | 1.0 | 0.42 |
| | SH | 98.55 (1.45) | 99.86 (0.14) | 1.0 | 0.50 |
| DAB | PEC | 95.93 (1.71) | 98.14 (0.92) | -1.56 | 0.26 |
| | SC | 97.97 (1.30) | 99.63 (0.37) | -1.29 | 0.29 |
| | SH | 100 (0.0) | 99.75 (0.25) | 1.0 | 0.39 |
| HW | PEC | 95.09 (2.63) | 95.43 (2.55) | -0.083 | 0.94 |
| | SC | 98.86 (1.14) | 92.54 (7.46) | 1.0 | 0.42 |
| | SH | 86.38 (6.76) | 93.99 (3.02) | -1.99 | 0.19 |

*p values are derived from paired sample t tests.

‡In the RCM, muscle relaxation in both 20 Hz frequency groups was 100%. Therefore, t statistics cannot be computed because the standard error difference is 0 and the groups are assumed to be indistinguishable (NS = not significant).

complete muscle fusion, such as when we gave 90 Hz and 100 Hz stimulation to the golden-collared manakin's SH. These data indicate that high contraction-relaxation cycling rates translate to corresponding high frequency wing oscillations.

## Species differences in forelimb muscle twitch speed dynamics

Next, to evaluate species differences in in situ contractile dynamics of the PEC, SC, and SH, we compared non-linear regression models that characterize muscle contraction speeds in response to varying stimulation frequencies (*Figures 3A,B and C*). For each muscle, models differed significantly among taxa (*Figure 3A,B and C*; PEC: $F_{16,130}$ = 2.50, p =0.0023; SC: $F_{16,140}$ = 5.50, p<0.001.; SH: SC: $F_{16,134}$ = 15.68, p<0.001). This suggests that there is performance variability in these muscles, and we suspect that such results underlie functional differences in the way in which the forelimb musculature controls rates of wing flapping.

## Evolution of an unusually fast wing muscle in golden-collared and red-capped manakins

Building on the analyses above, we investigated species differences in muscle contraction speeds that might explain the ability of certain taxa to rapidly move their limbs during display performances. For each muscle, we therefore compared the half-relaxation frequency among species (*Figures 3D,E and F*). This measure is the regression models' estimate of intrinsic contractile behavior, and it is defined as the maximum stimulation frequency at which the muscle's percent relaxation is half of its predicted functional range (see Methods for further details). We used this measure because it provides a consistent and unbiased metric by which we can compare muscle contraction speeds across birds. For both the PEC and the SC, half-relaxation frequencies averaged ≈50 Hz and were indistinguishable among species (*Figure 3D*, PEC: $F_{4,10}$ = 0.85, p = 0.53; *Figure 3E*, SC: $F_{4,11}$ = 1.74, p = 0.39). By contrast, in the SH, the half-relaxation frequencies differed significantly among taxa (*Figure 3F*, $F_{4,11}$ = 10.09, p = 0.0011), and post-hoc analysis of this effect showed these frequency measures were significantly higher in both golden-collared and red-capped manakins compared to

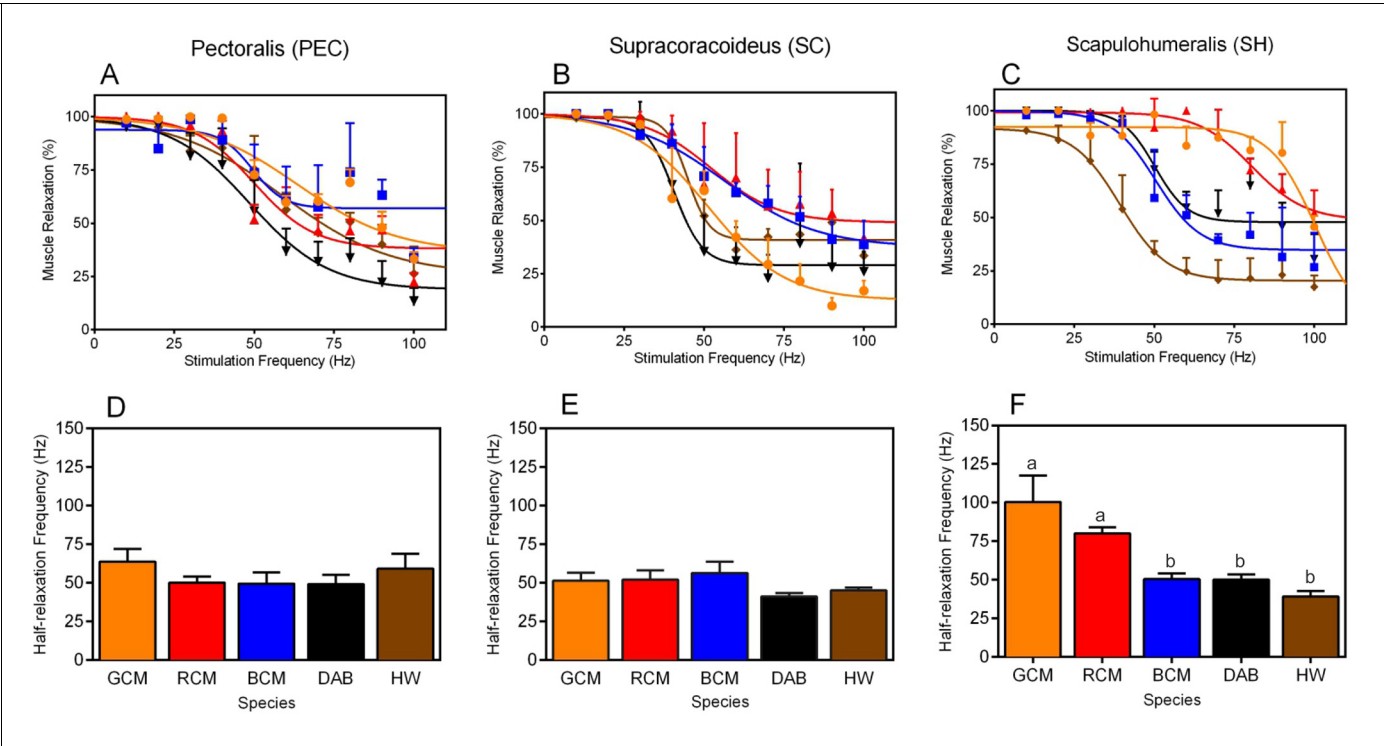

**Figure 3.** Muscle twitch speed dynamics in the (A, D) *pectoralis*, (B, E) *supracoracoideus*, and (C, F) *scapulohumeralis* of the five avian species included in our study. (A–C) Non-linear models generated to depict the relationship between mean muscle relaxations in response to different muscle stimulation frequencies. In each graph, muscle relaxation at a given stimulation frequency represents the mean (± 1 SEM) among individuals of a given species. (D–F) Half-relaxation frequencies of the different forelimb muscles across the five species in our analysis. Data represent mean (± 1 SEM) half-relaxation frequency for the given species. (F) For the SH, differences in the letters atop each bar denote statistically significant differences in mean half-relaxation values, according to post-hoc analyses (SNK tests, p<0.05). In all graphs, GCM = golden-collared manakin (orange); RCM = red-capped manakin (red); BCM = blue-crowned manakin (blue); DAB = dusky antbird (black); and HW = house wren (brown).

The following source data is available for figure 3:

**Source data 1.** Mean percent recoveries of the main wing muscles at different stimulation frequencies (see Materials and methods) across all species.

the other three species in the study (*Figure 3G* and *Table 2*; golden-collared: SNK post-hoc tests, q≥6.17, p<0.01; red-capped: SNK post-hoc tests, q≥3.64, p<0.05). Estimates of these two birds' SH half-relaxation frequencies were exceptionally high, coming in at ≈100 Hz and ≈80 Hz, respectively. Moreover, the SH half-relaxation frequencies between golden-collared and red-capped manakins were statistically indistinguishable (*Figure 3F* and *Table 2*; SNK post-hoc tests, q = 2.52, p>0.05).

Finally, we used our models to evaluate whether twitch speeds obtained from the golden-collared and red-capped manakin SH are in theory sufficient to drive wing oscillations necessary for each species' display. In the case of the golden-collared manakin, we found that the SH achieves on average ≈85% relaxation in response to stimulation frequencies of around 95 Hz. In the red-capped manakin, the SH achieves the same amount of mean relaxation in response to stimulation frequencies of roughly 70 Hz. The observation of concurrent wing movements at these stimulation frequencies (see above) shows that these changes in muscle length can actuate humeral retraction; thus, our results suggest that the SH is more than capable of driving the natural wing oscillations that make up each species' wing display (*Bostwick and Prum, 2003*; *Fusani et al., 2007*; *Fuxjager et al., 2013*). Such rapid kinetics of the SH suggest that this muscle features specializations that are similar to those of so-called 'superfast' muscles (*Rome et al., 1996*; *Elemans et al., 2008*; *Elemans et al., 2004*).

**Table 2.** Summary of SNK post-hoc results comparing SH half-relaxation frequencies between species, with statistically significant differences are shown in boldface typesetting. GCM = golden-collared manakin; RCM = red-capped manakins; BCM = blue-crowned manakin; DAB = dusky antbird; HW = house wren.

| Pairwise comparison | q statistic | DF | p value |
| --- | --- | --- | --- |
| GCM vs. RCM | 2.52 | 11 | >0.05 |
| **GCM vs. BCM** | **6.17** | **11** | **<0.01** |
| **GCM vs. DAB** | **6.65** | **11** | **<0.01** |
| **GCM vs. HW** | **7.58** | **11** | **<0.01** |
| **RCM vs. BCM** | **3.64** | **11** | **<0.05** |
| **RCM vs. DAB** | **3.96** | **11** | **<0.05** |
| **RCM vs. HW** | **5.06** | **11** | **<0.05** |
| BCM vs. DAB | 0.062 | 11 | >0.05 |
| BCM vs. HW | 1.42 | 11 | >0.05 |
| DAB vs. HW | 1.45 | 11 | >0.05 |

## Discussion

Our data provide the first insight into how animal motor systems are designed to perform spectacular behavioral displays. In both the golden-collared and red-capped manakin, the main humeral retractor muscle (the SH) shows half-relaxation frequencies of ≈80–100 Hz, while the humeral elevator (the SC) and the humeral depressor (PEC) show far lower half-relaxation frequencies (at ≈50 Hz). Importantly, these differences are not found in the other species within our analysis, including (*i*) closely related taxa that perform physically elaborate displays without rapid wing movements and (*ii*) taxa that inhabit similar flight environments. Altogether, these data uncover a close association between the evolution of rapid contractile kinetics of the SH and the ability to produce exceptionally fast wing movements as socio-sexual signals.

### A 'superfast' limb muscle

We hypothesize that the SH is one of the main actuators of the rapid forearm movements that are incorporated into golden-collared and red-capped manakin displays. Kinematic studies suggest that these species generate their wing sonations by moving their forelimbs at frequencies of 55–63 Hz and 45 Hz, respectively (*Bostwick and Prum, 2003*; *Fusani et al., 2007*; *Fuxjager et al., 2013*). Our results show that these rates fall well within the range of stimulation frequencies at which significant recovery of the SH is observed (at least 85%), and we find that individual stimulations are accompanied by observable wing movements. In addition, these same kinematic studies suggest that the bulk of the rapid forearm movement is driven by repetitive humeral retraction, and such movement of the wing is fully consistent with the biomechanical role of the SH (*Dial, 1992*; *Dial et al., 1991*). Thus, these data collectively suggest that the SH is the main muscle driving display production in both animals.

Since the elaborate displays of manakins are used for courtship and male-male rivalry (*Bostwick and Prum, 2003*; *McDonald et al., 2001*; *Barske et al., 2011*), sexual selection likely favored the emergence of rapid contractility in the SH. To this end, these superfast kinetics make the SH in these two birds the fastest vertebrate limb muscles currently known, at least with respect to measures of twitch contraction. Moreover, we know that these frequencies markedly exceed the typical wing-beat frequencies during flight (25–30 Hz) for birds of similar size (*Donovan et al., 2013*; *Pennycuick, 2001*), which again implies that sexual selection has positvely favored the emergence of an extreme muscle phenotype to accommodate the evolution of an equally extreme behavior.

With this in mind, it is interesting to speculate about why sexual selection has not forced faster display maneuvering, given that the SH appears capable of more rapid contractile kinetics than *roll-snapping* or *clapping* demands. Studies in golden-collared manakins suggest that females

preferentially mate with males that perform certain display maneuvers at fractions-of-a-second faster speeds (*Barske et al., 2011*; *2015*). Therefore, one intriguing possibility is that other wing muscles, like the 'slower' SC and PEC, also contribute to display production by helping position the wings (*Fusani et al., 2014*; *Schlinger et al., 2013*). In doing so, these muscles may limit the overall speed with which a given maneuver can be performed, owing to the effects of natural selection that presumably preserve these tissues' force-generating properties for powered flight (see below). If this is the case, then display speed itself may be constrained by neural circuits that integrate and/or coordinate the activation of numerous forearm motor units, as opposed to only the speed of the SH.

## Design of the motor system controlling behavior

Our results uncover a possible motor design that enables birds to produce rapid wing movements for a display, while preserving their ability to generate aerodynamic force needed to sustain powered flight. This, in effect, is rooted in our finding that muscle twitch speeds are modified in the SH, but not the SC or PEC. Rather, the latter two muscles maintain half-relaxation rates of ≈50 Hz, which are similar to the half-relaxation frequencies of all three muscles in the non-wing-displaying blue-crowned manakin, dusky antbird, and house wren. Thus, in golden-collared and red-capped manakins, the SC and PEC twitch speeds are largely conserved, which means that the ability of these muscles to generate aerodynamic force needed for powered flight is not compromised by the mutually exclusive ability to contract rapidly (*Rome et al., 1999*; *Young and Rome, 2001*). Flight in these two manakins is therefore unencumbered (*Moore et al., 2008*). To this end, this model is consistent with the biomechanical role of the SH, which contributes to aerodynamic force by adjusting the wings' angle of attack, area, and/or camber (*Dial, 1992*; *Dial et al., 1991*). The SH therefore likely contributes least to the generation of power for flight, and thus presents more opportunity for evolutionary modification without severely compromising flight ability. To our knowledge, this is the first glimpse into how evolution might circumvent trade-offs between muscle contraction speed and force generation to favor the emergence of rapid appendage displays, while maintaining normal locomotion.

Additional work is clearly needed to examine these ideas further, especially since the physiological and biomechanical mechanisms that underlie animal flight vary significantly among taxa. In vivo studies of muscle performance during display production would be especially helpful, as well as a better understanding of the force-velocity relationships of these birds' wing muscles.

## Putative 'costs' of a rapidly contracting wing muscle

The muscular design described above may impose a 'cost' on flight that has not yet been identified. Past work shows that both golden-collared and red-capped manakins are strong fliers, relative to other similarly sized tropical birds (*Moore et al., 2008*). This result indicates that dramatically enhancing the contraction speed of the SH does not altogether impede or diminish locomotion. However, studies have noted that both of these species produce a muffled fluttering sound when they fly (*Bostwick and Prum, 2003*). It is unclear if this sound, like the other sonations these birds produce, is a sexual signal, or whether it is a by-product of physiological and/or biomechanical adaptations that enable elaborate courtship maneuvering. If the latter is true, then our results might provide a mechanism for such action, as the ability of the SH to appropriately control wing positioning (*Dial, 1992*) might be altered by this muscle's performance shift. With this in mind, it will be interesting to pursue putative 'costs' of sacrificing force for speed in the SH, but not the SC and PEC.

## Conclusion

Our results provide the first insight into how the complex motor system for flight is designed to support wing-based acrobatic and elaborate behavioral displays. Specifically, in two species that use rapid wing movements as part of an adaptive behavioral display, we show that one of the muscles actuating wing movement has evolved 'superfast' contraction kinetics, while the kinetics of other muscles that provide more power for flight are functionally conserved. Thus, our study provides evidence not only for the emergence of the fastest known vertebrate limb muscle, but also a unique evolutionary design of the forelimb muscular system that enables *both* rapid movement for displaying and force-generating movement for locomotion.

## Materials and methods

### Animals

Adult male manakins were captured via passive mist netting at their respective breeding leks (golden-collared manakins, $n = 3$, red-capped manakins, $n = 3$, blue crowned manakins, $n = 3$). At the same time, adult male dusky antbirds ($n = 4$) and house wrens ($n = 3$) were captured from their breeding territories by luring individuals into the net with conspecific playback. Birds were immediately transported to our nearby laboratory for muscle recordings (see below), after which they were euthanized with an overdose of isoflurane and their tissues were removed for another study. We inspected the testes of each specimen to verify that they were enlarged and that the individual was in reproductive condition. The work was conducted from March to April in Gamboa, Panama at the Smithsonian Tropical Research Institute (STRI). All appropriate institution and governmental authorities approved the methods described herein, including the relevant Institutional Animal Care and Use Committees (IACUC) from STRI and Wake Forest University.

### Experimental design

Using a similar approach described elsewhere (*Elemans et al., 2004*; *2008*), we assessed the twitch speed dynamics of the PEC, SC, and SH (*Figure 1B*) in each individual collected from the field. Contraction-relaxation speed within a stimulus period was assessed by measuring the degree to which each muscle relaxed relative to its un-stimulated length (i.e., percent recovery) in response to different frequencies of electrical stimulation (see *Figure 2A*). All muscles were subjected to the following stimulation frequencies: 10 Hz, 20 Hz, 30 Hz, 40 Hz, 50 Hz, 60 Hz, 70 Hz, 80 Hz, 90 Hz, and 100 Hz. Each stimulation train consisted of 10 separate pulses. Pulse duration was set to 1 ms, and the pulse current was set between 0.5–0.8 mA. Stimulation trains were always delivered from low frequency (10 Hz) to high frequency (100 Hz). We space subsequent stimulation trains 1 min apart from each other to allow the muscle to temporarily rest.

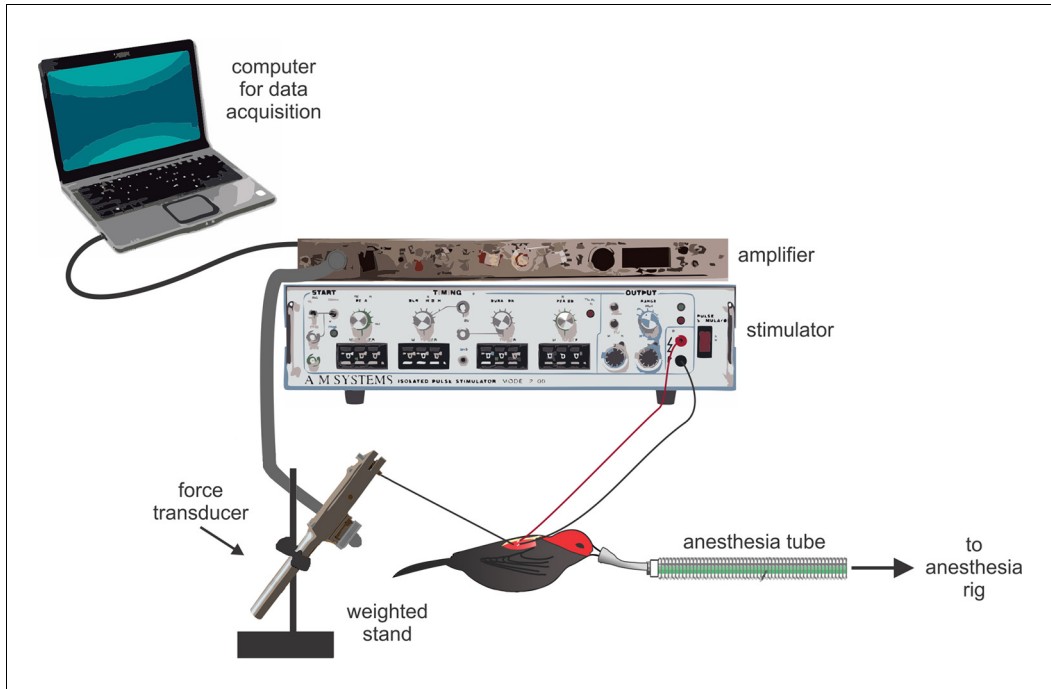

**Figure 4.** Schematic representation of the experimental set up used to record muscle twitch in situ. Birds were deeply anesthetized with isoflurane; their muscles were exposed, attached to the force transducer, and implanted with silver electrode wires. Data were collected on a nearby laptop computer. Note that the elements in this figure are not drawn to scale.

To confirm that our results were not confounded by damage or exhaustion of the muscle in response to repeated stimulations, we delivered a second 20 Hz stimulation train 1 min after the series' final 100 Hz stimulation (*Figure 2A*). We then compared the muscle recoveries in response to these two (temporally spaced) 20 Hz stimulations. In this study, every muscle was subjected to a single experimental series (i.e., 10 Hz to 100 Hz pulse as described above), and thus each muscle was also subjected to this validation procedure.

## Muscle preparations

Muscle twitch speeds were recorded in situ (*Figure 4*). For all preparations, birds were restrained on a soft foam pad that was securely attached to the surgical bench. Birds were then anesthetized with isoflurane (2–4% in $O_2$). To prepare the PEC and SH, we first cut a small (1 cm) incision in the skin. We then exposed the moist surface of the muscle by gently pushing apart the skin. Once the muscle was in clear view, we implanted it with the stripped ends (1–2 mm) of two insulated silver wire electrodes (diameter: 0.14 mm). These electrodes were then connected to a nearby stimulator (Model 2100, A-M Systems, WA, USA). We next fastened a stainless steel hook (0.1 mm diameter) to the muscle directly adjacent to the implanted electrodes. This hook was connected through a monofilament line to a force transducer (Model FT03, Grass Technology, RI, USA), which was tightly clamped to a heavy stand (≈6 kg). As soon as this preparation was completed, we placed a drop of normal avian saline (0.9%) over the exposed muscles to prevent tissues desiccation during the recording session. We adjusted the slack in the line to the force transducer by slightly moving the heavy stand; this allowed us to maintain tension in the line between the muscle and force transducer, and thus optimize the sensitivity of muscle twitch recordings without overloading the force transducer signal. When the recordings were completed, the electrodes and monofilament line from the force transducer were removed from the muscle, and the incision was closed using Vetbond tissue adhesive. All surgical preparations occurred at room temperature, which was similar to the outside ambient temperature (≈30°C).

We prepared the SC in a similar manner to the PEC and SH; however, we made a few modifications to this surgery, given that the SC lies deep to the PEC and is more difficult to access (*Figure 1*). First, we cut a small (1.5 cm) incision in the skin above the furcula. We then gently moved aside the underlying fat-pad and exposed the inter-clavicular air-sac. To gain access to the SC, we carefully moved the membrane of this air-sac to the side, taking great caution not to puncture it. From this angle, we could then see the SC positioned above the keel and below the PEC. We quickly implanted the SC muscle with the stripped ends of the silver electrodes, and allowed the inter-clavicular air-sac to fall back into position. Finally, we gently moved the fat pad back over the furcula to prevent desiccation during the recordings and placed a small drop of normal avian saline over this tissue. Given the deep (and difficult) position of the SC, it was not possible to fasten the hook from the force transducer directly to the muscle for twitch speed recordings. Instead, we attached the hook to the bird's elbow, and recorded the action of the stimulated SC as movement of the wing. When recordings were completed, we lightly pulled the electrodes from the muscle, removed the force transducer line from the elbow, and closed the dissection with Vetbond glue.

Both the stimulator and force transducer were connected to a laptop through an A-D converter (Model NI USB-6212, National Instruments, TX, USA). The signal from the force transducer was first amplified (5K–10K) and low-pass filtered (3000 Hz) using an AC/DC strain gage amplifier (Model P122, Grass Technologies). We collected all recordings in AviSoft-RECORDER (v.4.2.22) and measured the data in Praat software (v.5.4.21, P. Boersma and D. Weenink). This was accomplished by measuring the percentage at which muscles relax after each stimulatory pulse within a given stimulation train. Complete muscle relaxation (100% relaxation) occurred when the extra tension detected by the force transducer in response to stimulation was fully relieved (i.e., the signal returned to its baseline level). Partial muscle relaxation occurred when the extra tension placed on the force transducer in response to stimulation was *less than* fully relieved (i.e., summation occurred). Inasmuch, the values of partial relaxation ranged between 99% and 0% (full fusion) and were calculated by dividing the actual amount of relaxation by the amount of relaxation that would otherwise be necessary for full recovery.

For each stimulation train, we averaged the percent recovery values for the first eight stimulations. This corresponds to similar numbers of wing-oscillations that golden-collared and red-capped manakins use in their forelimb displays (*Bostwick and Prum, 2003*; *Fuxjager et al., 2013*). Notably,

in our calculations, we used baseline as the measure of a fully relaxed (un-stimulated) muscle, even though summation changes peak force production over the course of the stimulation train. As such, saturation levels may not always reflect 0% recovery.

## Statistical analysis

For each muscle, we compared the percent recoveries obtained from 20 Hz stimulations administered at the onset and after the experimental series using paired *t*-tests. One exception to this analysis occurred in red-capped manakins, as 20 Hz stimulations elicited 100% recovery in all cases. Variation across biological replicates is therefore 0.0, which means that statistics cannot be performed. We interpret these results to indicate no significant difference in muscle recovery in response to 20 Hz stimulations before or after the experimental series, which is entirely consistent with the results of the paired *t*-tests obtained from the other species (see *Table 1*).

Next, we examined muscle twitch dynamics by fitting the data with a four-parameter logistic non-linear regression model. This type of model characterizes data that form a reverse sigmoidal curve, and it is widely used to assess a variety of biological functions [e.g., *Kent et al., 1972*, *Johnson et al., 2003*]. We constrained model thresholds at or below 100% (maximum possible percent relaxation) and model saturations at or above 0% (minimum possible percent recovery). Each model produced an estimated inflection point (± 1 SEM), which represents the muscle's half-relaxation frequency. This metric is defined as the stimulation frequency at which the muscle's average recovery equals half of its predicted functional range. The half-relaxation point is biologically significant, in that it represents a change in muscle length that is still highly effective in actuating movement of the appendage (wing). This was validated in all species by watching the wings move in response to such stimulations (also see Results). Thus, at the level of muscle fusions associated with the half-relaxation frequency, generation of the display behavior is still highly feasible. To this end, we statistically compared models and their estimates of half-relaxation frequency only within muscle type, considering that differences in the preparations of the separate muscles might affect model parameters. Therefore, we used extra sum-of-squares tests to compare models across species and one-way ANOVAs to compare half-relaxation frequencies across species. Significant effects from ANOVAs were followed by Student-Newman-Keuls (SNK) post-hoc comparisons to control for multiple pairwise contrasts.

## Data accessibility

Data used for our analyses are included as a supplemental file (*Figure 3—source data 1*).

# Acknowledgements

We thank STRI and its administrative staff for their assistance with this project, as well as the Autoridad Nacional del Ambiente and the Autoridad del Canal de Panama for permission to conduct this work. We thank Leonida Fusani and Barney Schlinger for logistical support with the field season, as well as Mike Ryan for providing laboratory space at STRI. Finally, we thank Kyla Davidoff for comments on this manuscript and Eric Schuppe for help with the figures. This research was supported by intramural start-up funds from Wake Forest University (to MJF) and a research grant from the American Ornithologists' Union (to MJF).

# Additional information

### Funding

| Funder | Author |
| --- | --- |
| Wake Forest University | Matthew J Fuxjager |
| American Ornithologists' Union | Matthew J Fuxjager |

The funders had no role in study design, data collection and interpretation, or the decision to submit the work for publication.

## Author contributions

MJF, FG, Conception and design, Acquisition of data, Analysis and interpretation of data, Drafting or revising the article; AD, GDS, SG, Acquisition of data, Analysis and interpretation of data, Drafting or revising the article

## Author ORCIDs

Matthew J Fuxjager, http://orcid.org/0000-0003-0591-6854

## Ethics

Animal experimentation: This study was performed according to the protocol approved by the Institutional Animal Care and Use Committees (IACUC) of Wake Forest University (A13-155) and the Smithsonian Tropical Research Institute (2014-0101-2017-2-A2). Animal collections were conducted with the approval of the Government of Panama and the Autoridad Nacional del Ambiente and the Autoridad del Canal de Panama.

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
