## [Decision Letter]

Thank you for submitting your work entitled "Select forelimb muscles have evolved superfast contractile speed to support acrobatic social displays" for consideration by *eLife*. Your article has been favorably evaluated by Eve Marder (Senior editor) and two reviewers, one of whom, Russell Fernald, is a member of our Board of Reviewing Editors. The other reviewer, David Lentink, has also agreed to share his name.

The reviewers have discussed the reviews with one another and the Reviewing Editor has drafted this decision to help you prepare a revised submission.

This exciting and original manuscript presents data identifying high speed muscular contraction associated with social displays in some species of acrobatic manakins. Specifically, the authors identify one specific forelimb muscle to contract at superfast speeds and claim that this is now the fastest limb muscle known in any vertebrate. The experiments are straightforward and well presented.

Essential revisions:

Please provide more detail on how the experiment was conducted and make all data available. It would be helpful to add a clear diagram that explains the in vitro experiments in addition to more detail on the surgical procedures so this work can be replicated and extended straight-forward by others. Please include figure with a diagram of the SH muscle.

---

## [Author Response]

*Please provide more detail on how the experiment was conducted and make all data available. It would be helpful to add a clear diagram that explains the* in vitro *experiments in addition to more detail on the surgical procedures so this work can be replicated and extended straight-forward by others. Please include figure with a diagram of the SH muscle.*

We appreciate these comments, and we are happy to add more experimental details to make our methods more thorough and replicable. We have therefore made the following changes:

1) We have added more methodological detail, particularly with respect to our surgical procedures. In this regard, please note that our muscle recordings were performed in situ, such that the muscle was never removed from the bird. Our methodology is relatively more straightforward than many studies that collect muscle twitch speeds in vitro. We realize that fact, which is subtle but important, may have been lost in the paper, and we therefore have added statements in the Introduction (third paragraph) and Results (subsections “Validation of in situ muscle twitch recordings” and “Species differences in forelimb muscle twitch speed dynamics”) that specifically state that muscle recordings were collected in situ.

2) We have included a supplementary file that shows our study’s source data.

3) We modified Figure 1. It now shows the species we use in our study, as well as a schematic of the main wing muscles (PEC, SC, and SH) and where they are located within the bird. To this end, we also moved the schematic of our experimental work-flow and the exemplar twitch recordings to a new figure (i.e., Figure 2) by themselves.

4) Finally, in our Methods, we have drawn a new figure that illustrates our experimental set up to help readers better visualize how we recorded muscle twitch in situ.